# Predictors and Early Markers of Response to Biological Therapies in Inflammatory Bowel Diseases

**DOI:** 10.3390/jcm10040853

**Published:** 2021-02-19

**Authors:** Giuseppe Privitera, Daniela Pugliese, Gian Ludovico Rapaccini, Antonio Gasbarrini, Alessandro Armuzzi, Luisa Guidi

**Affiliations:** 1Dipartimento Universitario di Medicina e Chirurgia Traslazionale, Università Cattolica del Sacro Cuore, 00168 Roma, Italy; gpp.privitera94@gmail.com (G.P.); gianludovico.rapaccini@policlinicogemelli.it (G.L.R.); antonio.gasbarrini@policlinicogemelli.it (A.G.); alessandro.armuzzi@policlinicogemelli.it (A.A.); 2CEMAD—IBD UNIT—Dipartimento di Scienze Mediche e Chirurgiche, Fondazione Policlinico Universitario A. Gemelli IRCCS, 00168 Roma, Italy; daniela.pugliese@policlinicogemelli.it

**Keywords:** inflammatory bowel disease, biological therapy, predictors, biomarkers

## Abstract

Inflammatory bowel diseases (IBD) are chronic conditions that primarily affect the gastrointestinal tract, with a complex pathogenesis; they are characterized by a significant heterogeneity of clinical presentations and of inflammatory pathways that sustain intestinal damage. After the introduction of the first biological therapies, the pipeline of therapies for IBD has been constantly expanding, and a significant number of new molecules is expected in the next few years. Evidence from clinical trials and real-life experiences has taught us that up to 40% of patients do not respond to a specific drug. Unfortunately, to date, clinicians lack a valid tool that can predict each patient’s response to therapies and that could help them in choosing what drug to administer. Several candidate biomarkers have been investigated so far, with conflicting results: clinical, genetic, immunological, pharmacokinetic and microbial markers have been tested, but no ideal marker has been identified so far. Based on recent evidence, multiparametric models seemingly hold the greatest potential for predicting response to therapy. In this narrative review, we aim to summarize the current knowledge on predictors and early markers of response to biological therapies in IBD.

## 1. Introduction

Crohn’s disease (CD) and ulcerative colitis (UC) represent the two main forms of inflammatory bowel diseases (IBD). These are chronic conditions with a relapsing-remitting course that affect more than 5 million people worldwide, mostly in Western countries; their epidemiology is rapidly changing, with a sharp increase in incidence in Western countries registered in the previous decades [1]. IBD carries a significant direct and indirect health-care burden, which is mainly represented by drugs like biologics and small molecules, but also includes hospitalization-related costs and days of absence from work due to the disease [2]. In this scenario, there is an urgent need for a more effective approach than today’s trial and error method, when it comes to starting therapies.

To date, IBD treatment is based on corticosteroids (for the acute phases), mesalamine (only for UC), traditional immunosuppressants and targeted therapies; this last category includes: anti-tumor necrosis factor α (TNFα), anti-integrin, anti-interleukin (IL) 12/23 and Janus kinases (JAK) inhibitors [3,4,5]. It has been shown that different pathogenic pathways can sustain bowel damage in IBD [6,7], so that two patients with similar clinical phenotypes can have different inflammatory pathways activated and, thus, respond to different targeted therapies. There is also evidence that, within the same patient, the immune system can exhibit a significant plasticity and change the inflammatory pathways that are activated during the course of disease [8]. Such a complexity can easily explain why current therapies are only of limited efficacy. While the armamentarium for IBD treatment is constantly expanding—with new drugs targeting different pathogenetic pathways—there is still a significant proportion of patients who do not respond to therapy: data from clinical trials and real life report clinical efficacy of a single drug in up to 60% of patients [9,10]. Whether these patients would respond to another agent is not possible to foretell, but there is strong evidence that second- and third-line agents can be effective even in the case of primary failure, although to a lesser extent [11]. Furthermore, a substantial percentage of patients experience secondary failures [12]: in these cases, unless surgery is made mandatory by disease complications, the usual choice is to try another medical treatment.

Taken together, these considerations point to the necessity for the development of new prognostic tools able to identify those patients who would benefit from an early introduction of advanced therapies, to predict their response to a specific therapy and to assess response at an early point in treatment. Some clinical features have been identified so far, but they mainly identify patients who would most benefit from immunosuppressive therapies and/or patients who are more likely to respond to medical treatment, while they do not offer much information about patients’ response to a specific drug. Implementation of personalized medicine into IBD routine management represents one of the most compelling challenges of coming years, in order to provide patients with better clinical care in parallel with a reduction of costs for the health-care systems.

In this narrative review, we aim to summarize current knowledge regarding predictors and early biomarkers of response to biological therapies in IBD patients (Figure 1).

## 2. Traditional Markers

Associations of patients’ and disease characteristics with response to therapy have been widely investigated, but results have been generally discouraging. Age, gender, weight and smoking status have not been confirmed to correlate with response to anti-TNF or other targeted agents [13]. Two recent meta-analysis suggested that early treatment of CD is associated with better response rates [14,15], and one of those also observed an association with higher rates of mucosal healing [15]; however, no differences between different drugs have been observed so far. In CD patients, disease location has not been found to be associated with treatment outcomes of anti-TNF and ustekinumab [13]; of note, one study reported an association between colonic localization and better responses to vedolizumab [16], that was not confirmed in others. Some studies have reported a correlation, in CD patients, between inflammatory phenotype and better response rates with TNF antagonists, compared to stricturing or penetrating diseases [17,18,19,20]; such findings have not been confirmed for vedolizumab and ustekinumab.

Various serological biomarkers have been investigated for their potential predictive role. In a recent retrospective study on elderly (>60 years old) patients, a higher serum triiodothyronine-to-thyroxine (T3/T4) ratio was found to be associated to mucosal healing, regardless of the biological drug used [21]. C-reactive protein (CRP) levels haves been inconsistently associated with response to biological therapies. A correlation between higher CRP levels and better response to TNF antagonists has been reported in CD patients [22,23,24,25]; conversely, a negative correlation between CRP levels and response rates to TNF inhibitors has been observed in UC [26,27,28]. Similar findings have also been observed with vedolizumab, but not with ustekinumab [13]. However, such results have not been confirmed by many other studies. Higher CRP levels can help identify those patients whose symptoms are actually dependent on active IBD and, in CD, can help discriminate inflammatory vs. stricturing phenotypes; on the other hand, such high levels can also be expression of a higher inflammatory burden, that is comprehensively associated with poorer response to medical treatment. Faecal calprotectin has been tested as a potential predictor of response, with disappointing results. Of note, in a prospective observational study it has been found that lower post-induction calprotectin levels were able to predict sustained clinical response and mucosal healing in IBD patients receiving anti-TNF treatment [29].

Previous exposure to biologics has been associated with poorer response to subsequent lines of therapies. Reason for discontinuation seems to have an impact on the likelihood of responding to second-line therapies: a Spanish retrospective study and 2 meta-analysis concluded that discontinuation due to anti-TNF intolerance was associated with higher rates of response to both second anti-TNF or other biologic agents [11,30,31]. Primary non-response to TNF antagonists seems to correlate with an even lower likelihood of response to second-line therapies, when compared to secondary loss of response (LOR) [30]. A retrospective study on UC patients showed that, in case of primary failure, out-of-class swap seems to be superior to in-class switch with regard to rates of clinical response and remission [32]. However, such findings have not been consistently confirmed in literature.

## 3. Genetic Markers

More than 240 susceptibility loci for IBD have been identified so far [33]. Such genes have greatly contributed to the understanding of IBD pathogenesis and to the identification of novel therapeutic targets. However, genetic markers have usually performed quite poorly in predicting response to a specific drug [34]. In Table 1, there is an overview of the studies investigating the association between genetic markers and response to therapy.

Genome-wide association studies have reported that disease susceptibility loci do not seem to substantially contribute to anti-TNF non-response. For instance, polymorphisms in the genes encoding for TNF or molecules involved in the TNF receptor pathway have been inconsistently associated with treatment response. A 2013 meta-analysis reported an association between 2 polymorphisms in the TNF promoter and responsiveness to TNF inhibition in IBD patients: specifically, the more common alleles were associated with better response rates [35]. Another meta-analysis found a positive correlation between polymorphisms in *FCGR3A*, *TLR4*, *TNFRSF1A*, *IFNG*, *IL6*, *IL1B* genes and better clinical response, whereas variants of TLR2 and TLR9 were negatively correlated [34]. Polymorphism in the Nuclear Factor kappa-light-chain-enhancer of activated B cells (NFkB) pathway, TNF pathway and pathways of other cytokines have been linked to treatment response in IBD patients treated with anti-TNF agents [36,49]. IL23 receptor polymorphisms have been associated with response to infliximab in UC patients [37]. Moreover, the HLA-DQA1*05, the HLA-DRB1 allele and polymorphisms at the FCGR3A locus (encoding IgG Fc receptor IIIa) have been correlated with an increased risk of anti-drug antibodies (ADA) formation in CD patients treated with anti-TNF agents [38,39,40]. Being large complex proteins, monoclonal antibodies—especially infliximab, that is a chimeric antibody—can stimulate the production of ADA, which are associated with treatment inefficacy [50]. Identifying patients who are more likely to develop ADA would be of great help, as we know that concomitant immunosuppression (with thiopurines and methotrexate) reduces the risk of their formation [51].

Another marker previously identified by gene array studies in mucosal biopsies of IBD patients is the IL13 receptor alpha 2 (IL13RA2) [52]. This biomarker has been more recently evaluated as mRNA expression in the mucosa of IBD patients prior to therapy and found to be specifically predictive of non-response to anti-TNF in terms of mucosal healing at 6 months with an area under the receiver operating characteristic (AUROC)of 0.90 for infliximab and 0.94 for adalimumab, *p* < 0.001 [53].

The NOD2 gene is associated with CD susceptibility and with a more aggressive course of disease [54,55]; it encodes for a protein that plays a role in eliciting the immune response and is implicated in the inflammatory pathway of TNF. Some studies have found an association between NOD2 variants and worse response to anti-TNF therapy [41,42,43]. Polymorphisms in the ATG16L1 gene have been associated better response rates [44] and longer benefit [46] in CD patients treated with TNF antagonists. An apoptotic pharmacogenetic index (API) has been proposed to predict treatment response to anti-TNF in CD patients. The index was based on single nucleotide polymorphisms of 3 apoptotic genes (Fas, Fas-ligand and caspase-9). The authors elaborated a model, combining API with clinical features, that was able to predict response to infliximab in luminal and penetrating CD [45]. Predictive models combining clinical and genetic features have been shown to be superior to models based on clinical characteristics only for predicting primary non-response to anti-TNF agents in CD (Area Under the Receiver Operating Characteristics [AUROC] 0.93 vs. 0.70, *p* < 0.0001) [46], UC (AUROC 0.87 vs. 0.57, *p* < 0.0001) vs. [47] and IBD patients (AUROC 0.89 vs. 0.72, *p* < 0.0001) [48].

Data on genetic variants associated with response to anti-integrin is scarce. In the phase 2 trial of etrolizumab (anti-β7 integrin) in UC patients, αE gene expression was found to be predictive of clinical remission at week 10 [56]. This result was subsequently enhanced by the finding that also higher levels of Granzyme A (which is highly expressed by αE+ cells) mRNA in colonic biopsies taken at baseline could identify UC patients in clinical remission at week 10 of etrolizumab therapy [57]. No association between genetic markers and anti-IL12/23 response has been identified so far.

## 4. Other Markers: Transcriptomics, Proteomics and Immunological Markers

Transcriptomics studies have provided some more pleasing results, suggesting that therapy response seems to rely on differential expression of significant genes, more than on genetic variants.

A putative biomarker identified by transcriptomics studies is the triggering receptor expressed on myeloid cells 1 (TREM-1), although with some discordant findings: a study of gene expression described a down-regulation in whole blood of non-responders to anti-TNF. However, these patients showed an up-regulation of TREM-1 and of chemokine receptor type 2 (CCR2)–chemokine ligand 7 (CCL7) in intestinal biopsies. In the same study, plasma cell frequencies were examined in intestinal biopsies by CD138+ staining and were considered able to predict anti-TNF response, being higher in non-responders (*p* = 0.0005). [58] A different study analyzed the expression of TREM1 in whole blood and in mucosal tissue and as protein level in the serum and found a significant reduction in IBD patients who achieved mucosal healing [59]. This pathway seems to be specific for anti-TNF response as no modifications was detected in patients treated with vedolizumab or ustekinumab.

Measures of TNF production have been studied as putative biomarkers of response to anti-TNF. In vivo imaging by endomicroscopy revealed higher numbers of mTNF+ cells in short-term (12 weeks) responders, after local fluorescent adalimumab administration [60]. A recent study analyzed in 42 IBD patients the in vitro production of TNF from peripheral blood mononuclear cells (PBMC) stimulated with lipopolysaccharide (LPS) and found it to be predictive of clinical response after 6 weeks of infliximab therapy. A cutoff of 500 pg/mL was identified in CD for short-term response with 100% sensitivity and 82% specificity [61].

High expression of a member of the IL6 family, oncostatin M (OSM), in the intestinal mucosa was found to predictive of refractoriness to anti-TNF therapy. The clinical response was assessed at week 8 and 30 in a cohort of patients treated with infliximab (from ACT1/2 studies) and at 6 weeks in a cohort of patients treated with golimumab (from the Program of Ulcerative Colitis Research Studies Utilizing an Investigational Treatment−Subcutaneous, PURSUIT study) [62]. This biomarker was also studied at baseline in serum and found to be predictive of mucosal healing at 54 weeks of infliximab treatment with an AUC of 0.91 in a study on 45 CD patients [63].

Assays of α4β7 occupancy in peripheral blood T cells showed almost complete blocking of this integrin in patients after vedolizumab treatment, irrespective of clinical response and also of circulating drug levels [64]. However, in a small study, vedolizumab responders had higher pre-treatment α4β7 expression on T effector memory cells (*p* = 0.0009 for CD4 and.0043 for CD8) and on natural killer (NK) cells (*p* = 0.0047) [65]. These results were partly confirmed at the tissue level by a preliminary study with confocal endomicroscopy with fluorescein isothiocyanate (FITC) labelled vedolizumab, where only CD patients were responders to subsequent therapy with the anti-integrin showed α4β7+ cells in the mucosa [66]. A study examined the in vitro assay of baseline peripheral blood CD4+ cells dynamic adhesion to recombinant MAdCAM-1 and the decrease of this effect by vedolizumab: these parameters have been suggested as predictors of clinical response at 15 weeks in a study on 21 UC patients [67]. Another small study explored serum markers of response in anti-TNF refractory patients, before starting vedolizumab. They found higher levels of IL6 in IBD patients who were subsequently non-responder, of sCD40L in CD non-responder patients and higher osteocalcin levels in UC responders [68]. More recently serum IL6 and IL8 measured at baseline and at week 10 of vedolizumab treatment were suggested as early markers of clinical response at 12 months [69]. The prognostic value of serum biomarkers measured at baseline and at early intervals during vedolizumab therapy was also explored in two studies. In UC, an increase of s-α4β7 an a decrease of s-MAdCAM-1, s-VCAM-1, s-ICAM-1, and s-TNF were found in clinical and endoscopic remitters at 26 weeks [70]. In CD patients, higher early serum levels of s-ICAM-1 and s-VCAM-1 and lower values of s-α4β7 were found in endoscopic remitters [71].

Limited data of transcriptomic and immunologic predictive biomarkers of response to IL23 inhibition are available. IL22 exerts a role of positive regulation on IL23 signalling. In the phase 2a trial of brazikumab (anti-IL 23) in CD patients, higher concentration of IL22 at baseline had an association with increased likelihood of response, even though the association did not test statistically significant [72].

Results from transcriptomic and immunologic studies are resumed in Table 2.

## 5. Therapeutic Drug Monitoring

Therapeutic drug monitoring (TDM) consists in the dosage of drug trough levels (TLs) and ADA titres. To date, reactive TDM (performed in patients with active intestinal symptoms) is recommended for the management of LOR to anti-TNF agents, in particular with infliximab and adalimumab [73]. Furthermore, it has been shown that a tailored management—based on reactive TDM—in the case of secondary non-response is more cost-effective than symptom-based dose escalation [74,75].

An association between higher TLs and better clinical outcomes has been observed in retrospective studies and post-hoc analysis of RCTs (Table 3) [76,77,78,79,80,81,82,83,84,85,86]. The PANTS study, where anti-TNF naïve patients with luminal CD were enrolled at the time of anti-TNF initiation and prospectively observed until treatment discontinuation, reported that low TLs of infliximab (<7 mg/L) or adalimumab (<12 mg/L) at week 14 correlated with primary non-response at week 14 (odds ratio [OR] 0.35, 95% confidence interval [CI] 0.20–0.62, *p* = 0.00038 for infliximab; OR 0.13, 95% CI 0.06–0.28, *p* < 0.0001 for adalimumab) and non-remission at week 54 (OR 0.29, 95% CI 0.16–0.52, *p* < 0.0001 for infliximab; OR 0.03, 95% CI 0.10–0.12, *p* < 0.0001 for adalimumab) [87]. Similar associations between TLs and therapeutic outcomes have also been observed for golimumab, both in the post-hoc analysis of the registration trial [88] and in some observational studies [89,90,91,92]. Some studies have suggested that specific disease phenotypes might need higher TLs to be adequately controlled. For instance, it has been proposed that, in patients with perianal fistulizing CD, higher than usual infliximab TLs are required to induce healing [93,94,95]. It has also been suggested that patients with acute severe ulcerative colitis (ASUC) might benefit from an optimized induction regimen (with an additional infusion at week 1), even though results have been inconsistent so far [96,97,98]. Furthermore, there is evidence that therapeutic levels depend also on the target considered: more ambitious targets might require higher TLs [99,100].

In the case of LOR to infliximab or adalimumab, TDM can predict the outcomes of subsequent therapeutic interventions and, therefore, help in choosing the correct management. In a retrospective work, Yanai et al. [115] investigated the outcomes of different interventions in patients experiencing LOR to infliximab or adalimumab, according to TLs and ADA. Patients with low TLs and negative ADA were the most likely to respond to therapy escalation. Out-of-class switch in patients with adequate TLs (>3.8 µg/mL for infliximab and >4.5 µg/mL for adalimumab) was associated with an increased likelihood of early response compared to therapy optimization (OR 7.8; 95% CI, 1.5–42; *p* = 0.02) and a trend towards longer duration of therapy response (log rank test *p* = 0.09); conversely, in patients with high titres of antibodies, switching to another anti-TNF was associated with longer duration of response (log rank test *p* = 0.03), compared to therapy optimization [115]. In a 2018 work, Roblin and colleagues found that pharmacokinetic status upon first anti-TNF failure could help predicting the outcome of second-line anti-TNF treatment: patients with adequate TLs at LOR were significantly less likely to achieve clinical remission at week 30, compared to patients with subtherapeutic or undetectable TLs, regardless of ADA titre (*p* ≤ 0.05 against each group). Notably, at week 102, patients with undetectable TLs and negative ADA and patients with subtherapeutic TLs showed higher rates of clinical remission, compared to those with therapeutic TLs or undetectable TLs and positive ADA; furthermore, patients with undetectable TLs and positive ADA at LOR were the most likely to develop ADA against the second anti-TNF [116]. These observations are in line with some more recent findings from the same group: in IBD patients switched to a second-line anti-TNF therapy due to immunogenicity, the addition of an immunomodulator seems to prevent ADA formation and to be associated with lower rates of clinical failure [117].

Using TDM proactively—to manage stable patients and optimize therapy based on TDM results—has not been shown to provide significant benefits. The Trough Concentration Adapted Infliximab Treatment (TAXIT) and the randomized controlled trial investigating tailored treatment with infliximab for active luminal Crohn’s disease (TAILORIX) RCTs, testing a proactive TDM approach in IBD patients, both failed to meet their primary endpoint and demonstrate the efficacy of proactive TDM for IBD management [118,119]. Conversely, in a recent RCT on 78 biologic-naïve paediatric CD patients receiving adalimumab, proactive TDM was found to be superior to reactive TDM for maintaining sustained steroid-free clinical remission from week 8 to week 72 (82% vs. 48%, *p* = 0.002) [120]. Evidence is emerging that highlight an association between TLs during induction phase and treatment efficacy, thus suggesting a potential role of early proactive TDM for the prevention of primary non-response [121].

Data on TDM with vedolizumab and ustekinumab is scarcer. Both agents are have lower immunogenicity compared to anti-TNF [50], and the presence of ADA has not been consistently associated with therapy inefficacy in registration trials [122,123,124,125]. Early vedolizumab TLs (mainly at weeks 6 and 14) have been variously associated with clinical remission [101,102], treatment persistence [103], biochemical remission [103,104,105], endoscopic response [126], mucosal healing [104,105] and even histological healing [106]; due to the heterogeneity of assays used, timepoints assessed and outcomes investigated, different threshold concentrations have been proposed. An association between vedolizumab TLs during maintenance periods and treatment outcomes has been reported in multiple observational studies [102,106,107]. Interestingly, in a retrospective analysis of IBD patients on maintenance therapy with vedolizumab, TLs <7.4 µg/mL, in patients experiencing LOR was associated with a significantly increased likelihood to respond to dose escalation compared to higher concentrations (OR 3.7, 95% CI 1.1–13, *p* = 0.04) [108]. A similar exposure-efficacy relationship has been observed for ustekinumab: early TLs (week 4 and 8) have been associated with clinical remission [109,110], decrease of faecal calprotectin [111] and biochemical remission [112], while TLs during maintenance have been correlated with clinical remission [109,110], endoscopic response [111,113] and need for optimization [114]. In a recent review, Alsoud et al. [127] proposed target TLs for vedolizumab and ustekinumab. During induction, TLs >20 µg/mL at week 6 and >14 µg/mL at week 14 have been suggested for vedolizumab, whereas for ustekinumab >14 µg/mL and >4 µg/mL at weeks 4 and 8, respectively; during maintenance phase, TLs >12 µg/mL for vedolizumab and >2.5 µg/mL for ustekinumab have been proposed [127].

In conclusion, TDM can prove extremely useful to inform the management of patients with loss of response. In case of secondary failure, objective confirmation of disease activity (preferably with endoscopy, or with biomarkers) should be quickly obtained [73], and endoscopic re-evaluation should be performed within 3-6 months after therapy adjustment [128]. In consideration of the well-established dose-response relationship observed with biologics, loss of response is frequently managed via therapy optimization (i.e.: dose-escalation or interval shortening) [76]. However, pharmacokinetic failure—which is therapy inefficacy due to inadequate TLs—only accounts for some cases of loss of response, and TDM can help to identify those patients who would benefit from optimization. Indeed, TDM is recommended to manage loss of response to anti-TNF agents, in order to discriminate patients who might need higher doses to achieve clinical remission from those who need to change therapy [73,76]. With regard to infliximab, both dose-escalation and interval shortening are considered valid options, and it has been previously demonstrated that either strategy is equally effective: in a retrospective study including 94 CD patients, shortening dose to 6 weeks was found to be as effective as doubling the dose to manage loss of response [129]. TDM for vedolizumab and ustekinumab has not been incorporated in routine clinical practice, yet; however, based on the evidence presented above, it is plausible that reactive TDM could contribute to identify patients who would benefit from vedolizumab or ustekinumab optimization. 

## 6. Gut Microbiota

While it is well established that microbiota alterations play a role in IBD pathogenesis, whether dysbiosis is a cause or rather a consequence of intestinal inflammation is not clear yet. Furthermore, beside alterations in microbiota compositions, it has been increasingly recognized that changes in microbiota metabolic functions have a meaningful impact on IBD pathophysiology [130]. Table 4 summarizes the main findings on the associations between microbiological features and response to therapy.

Different studies reported that microbial diversity at baseline, as well as higher abundance of specific taxa or species, could predict response to anti-TNF treatment [131,139,140]. It has been shown that anti-TNF treatment can alleviate intestinal dysbiosis in IBD patients, with enrichment in species diversity and an increased relative abundance of short-chain fatty acids (SCFAs) producing bacteria [132]. However, such an improvement has not been found to consistently correlate with treatment efficacy. It has also been shown that longitudinal changes in microbiota composition correlate with response to anti-TNF therapy: in particular, an increase in the Firmicutes phylum (most notably, F. prausnitzii) has been observed in responders [131,133,140].

A longitudinal prospective study on a paediatric cohort of IBD patients, who received immunomodulators or biologics, could not find an association between the degree of dysbiosis at baseline and treatment response, measured as mucosal healing after approximately one year; however, the authors observed significant differences, between responders and non-responders, in the relative abundance of specific microbial genera (namely *Akkermansia*, *Coprococcus*, *Fusobacterium, Veillonella, Faecalibacterium,* and *Adlercreutzia*), and found that baseline microbial data could predict treatment response with a 75.6% accuracy [134].

Zhuang and colleagues found that a prediction model, based on the combination of microbiota data, clinical activity and FCP at baseline, could predict infliximab response at week 30 (AUC = 0.938). In a recent work on biologic-naïve CD patients treated with infliximab, enrichment of Lachnospiraceaea and Blautia, after infliximab initiation, was found to be a predictor of treatment efficacy. In particular, Lachnospiraceaea and Blautia combined increase at week 6 was able to predict clinical remission at week 14 (AUC = 0.924, 95% CI 0.707–0.962; *p* < 0.0001), as well as clinical remission (area under the curve (AUC) 0.842, 95% CI 0.715–0.968; *p* < 0.001) and endoscopic response (AUC 0.891, 95% CI 0.789–0.993; *p* < 0.0001) at week 30 [135]. In a study by Wang et al., a positive correlation between sustained response and increase in SCFAs-producing bacteria was found in infliximab-treated paediatric IBD patients [141].

Aden and colleagues investigated the association between metabolic changes of gut microbiota and anti-TNF treatment in IBD patients. Via an in silico model, they showed that, at baseline, metabolic interchanges between microbes were more reduced in patients not achieving clinical remission; non-remitters were characterized by a significantly reduced number of mutualistic and antagonistic interactions between bacteria (which contribute to enhance the stability of microbial ecosystem) and by an increased number of competitive interactions (which reduce microbiota stability). Furthermore, they were able to identify bacterial stool metabolites that were associated with clinical remission at week 14: butyrate and its substrates were more frequently exchanged in patients achieving responding to anti-TNF therapy [136]. Ding et al. recently presented their results on metabolomics analysis on CD patients treated with anti-TNF agents, showing a distinct metabolic profile of serum, faecal and urinary metabolites that was able to discriminate between responders and non-responders at baseline. In particular, they identified higher levels of primary bile acids and lower levels of secondary and tertiary acids to be associated with non-response to anti-TNF treatment. Since bile acid metabolism is known to be dependent on specific intestinal bacteria, it has been advanced that the aforementioned differences between responders and non-responders might be related to differences in the gut microbiota [137].

In a study investigating the effect of infliximab therapy in IBD patients in clinical remission, it was found that infliximab TLs ≥ 5 µg/mL and MH were associated with increased bacterial diversity, richness and relative abundance of F. prausnitzii in gut microbiota at baseline [138]. Whether intestinal eubiosis should be considered a prognostic marker or might represent a therapeutic target associated with better clinical outcomes is yet to clarified. Ananthakrishnan and colleagues investigated the role of gut microbiome functions in predicting response to vedolizumab in IBD patients. They found that a predictive model incorporating clinical and microbial data (on composition and function) at baseline was able to predict clinical remission at week 14 (AUC 0.872); in particular, they found abundance of butyrate-producing bacteria and enrichment of branched chain amino acid biosynthesis pathways at baseline in remitters. Furthermore, they observed early changes in microbiota composition and functions, especially the reduction in oxidative stress pathways, in patients achieving remission [142]. Doherty et al. investigated gut microbiota in CD patients treated with ustekinumab. A predictive model based on clinical and microbiome data was able to predict response and remission at week 6 with 84.4% and 73.3% accuracy, respectively. They also observed enrichment in microbial diversity during ustekinumab therapy in responders, but not in non-responders [143].

## 7. Conclusions

The one size fits all paradigm for medical therapies has been strongly rejected by academics; however, we are still far behind in finding valid prognostic tool that can be implemented in real-life clinical practice. So far, the only tool of personalized medicine that has been widely incorporated in IBD clinical practice in regard to biotechnological therapies is represented by TDM for secondary non-response to infliximab and adalimumab.

Patients’ and disease’s markers probably have a role in predicting the course and aggressiveness of IBD in a certain patient, but have not proven sufficient to accurately foretell if a patient will respond to a specific therapy. Genetic and transcriptomic markers seemingly hold the potential to help predicting response to IBD therapies: a significant number of different loci has been investigated so far, but no ideal candidate for prediction of response has been identified. The susceptibility to ADA formation conferred by some genetic polymorphisms could prove extremely useful in real-life clinical practice, as it could help identify patients who might benefit from the association of an immunomodulators. With regards to TDM, whether higher TLs are causatively correlated to better clinical outcomes or are just a marker of decreasing inflammatory burden and treatment efficacy is still unknown. To date, there is not enough evidence to conclude that TLs should be considered a target per se. The immunological landscape of IBD is characterized by remarkable inter- and intra-patient variability and plasticity; thus, a greater comprehension of the immune pathways activated in a specific patient at a specific time-point—and the identification of reliable biomarkers thereof—could substantially contribute in tailoring the treatment for the patient. Finally, microbiological markers are undoubtedly attractive: gut microbiota seems to change in parallel with IBD phases and could, therefore, be regarded as dynamic, non-invasive biomarker that could inform about IBD activity and, potentially, predict response to therapy. Notably, most of recent studies seem to suggest that multiparametric models, incorporating different features, hold the highest predictive power. It is also worth mentioning that most of the findings discussed in this review represent just association, meaning that they need to be cautiously interpreted, as they do not prove causation. Indeed, it is of the utmost importance that predictive markers and models need to be validated in external cohorts, so as to prove their strength.

In conclusion, the implementation of personalized medicine represents one of the most crucial unmet needs in IBD. Given the growing expansion of the IBD population, the significant health-care costs associated with the disease and the expanding pipeline of therapies, future studies will need to look at the development of new prognostic tools that can enable us to choose the right drug, for the right patient, at the right time.

## Figures and Tables

**Figure 1 jcm-10-00853-f001:**
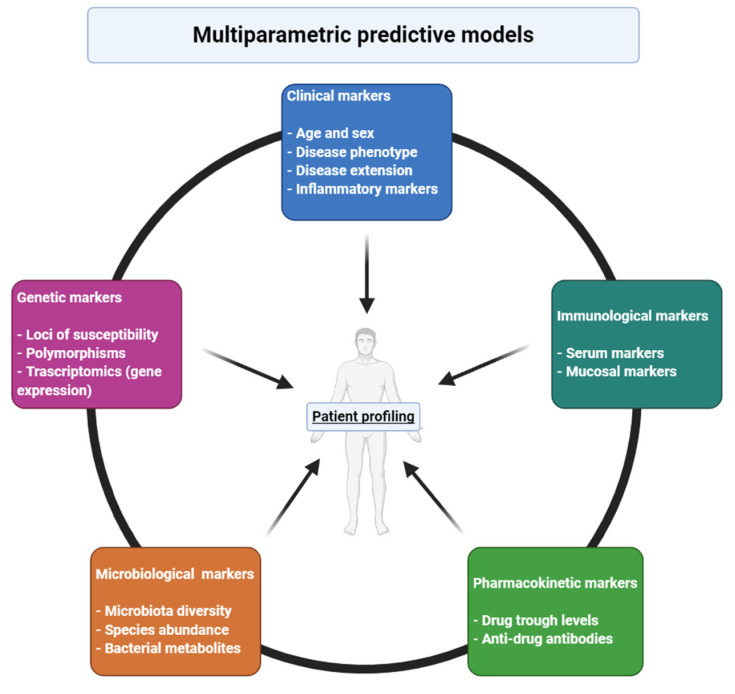
Different features that contribute to responsiveness to a certain drug and that needs to be included in multiparametric models for prediction of response in inflammatory bowel diseases. Created with BioRender.com.

**Table 1 jcm-10-00853-t001:** Genetic markers.

Study	Genetic Markers	Outcomes
Bek et al. 2016 [34]	*Polymorphisms in TLR2*, rs11938228, *TLR4*, *TLR9*, *TNFRSF1A*, *IFNG*, *IL6* and *IL1B* (rs4848306)	Clinical response to anti-TNF in IBD patients
Tong et al. 2013 [35]	Polymorphisms in TNF-α promoter (-308 A/G and -857 C/T)	Clinical response to anti-TNF in IBD e SpA patient
Bank et al. 2014 [36]	Polymorphisms implicated in NF-kB pathway: TLR2, TLR4, TLR9, LY96 (MD-2), CD14, MAP3K14 (NIK), TNFA, TNFRSF1A, TNFAIP3(A20), IL1B, IL1RN, IL6, IL17A, IFNG	Clinical response to anti-TNF in IBD patients
Jürgens et al. 2010 [37]	Polymorphisms in IL23R	Early response to infliximab in UC patients
Sazonovs et al. 2020 [38]	HLA-DQA1*05	Development of ADA against infliximab and adalimumab in CD patients
Billiet et al. 2015 [39]	HLA-DRB1	Development of ADA against infliximab in IBD patients
Louis et al. 2004 [40]	Polymorphism in IgG Fc receptor IIIa	Development of ADA against infliximab in CD patients
Niess et al. 2012 [41]	Polymorphisms in NOD2	Clinical response to anti-TNF in CD patients
Juanola et al. 2015 [42]	Polymorphisms in NOD2	Loss of response to anti-TNF in CD patients
Schäffler et al. 2018 [43]	Polymorphisms in NOD2	Lower anti-TNF TLs in CD patients
Koder et al. 2015 [44]	Polymorphisms in ATG16L1	Clinical response to adalimumab in CD patients
Hlavaty et al. 2007 [45]	Polymorphisms in Fas, Fas ligand and Caspase 9 (Apoptotic Pharmacogenetic Index)	Clinical response to infliximab in CD patients
Barber et al. 2016 [46]	Multiple polymorphisms (Combined clinical-genetic model)	Short- and long-term to anti-TNF in CD patients
Burke et al. 2018 [47]	Multiple polymorphisms (Combined clinical-genetic model)	Short- and long-term response to anti-TNF in UC patients
Wang et al. 2019 [48]	Polymorphisms in *TNFSF4/18*, *PLIN2*, rs762787, rs9572250, rs144256942, rs523781	Clinical response to anti-TNF in IBD patients

ADA: anti-drug antibodies; ATG16L1: autophagy-related 16 like 1; CD: Crohn’s disease; HLA: human leukocyte antigens; IBD: inflammatory bowel disease; IL: interleukin; MAP3K14: mitogen-activated protein kinase kinase kinase 14; NF-kB: nuclear factor kappa-light-chain-enhancer of activated B cells; NOD2: nucleotide-binding oligomerization domain-containing protein 2; PLIN2: perilipin 2; TLR: toll-like receptor; TNF: tumor necrosis factor; TNFR: tumor necrosis factor receptor; UC: ulcerative colitis; TNFSF: tumor necrosis factor superfamily; IFNG: interferon gamma.

**Table 2 jcm-10-00853-t002:** Immunological markers.

Study	Immunological Markers	Outcomes
Gaujoux et al. 2019 [58]	Higher expression of TREM-1 and CCR2-CCL7 in intestinal biopsies	Nonresponse to anti-TNF treatment
Verstockt et al. 2019 [53]	Lower expression of TREM-1 in whole blood and intestinal biopsies, lower concentration in serum	Mucosal healing in patients treated with anti-TNF
Atreya et al. 2014 [60]	Higher number of mTNF+ cells in intestinal biopsies	Short term (12 weeks) response to adalimumab
Jessen et al. 2020 [61]	TNF production > 500 pg/mL by PBMC stimulated with LPS	Clinical response to infliximab at week 6
West et al. 2017 [62]	Higher expression of OSM in intestinal biopsies	Refractoriness to infliximab (at weeks 8 and 30) and golimumab (at week 6)
Bertani et al. 2020 [63]	Lower serum concentration of OSM	Mucosal healing at week 54 in infliximab-treated patients
Boden et al. 2018 [65]	Higher expression of α4β7 on T effector memory cells and NK cells	Response to vedolizumab
Rath et al. 2017 [66]	Presence of α4β7+ cells in intestinal mucosa	Response to anti-integrin therapy
Allner et al. 2020 [67]	Higher dynamic adhesion of peripheral blood CD4^+^ T cells to MAdCAM-1 and more pronounced reduction of adhesion following treatment	Clinical response to vedolizumab in UC patients
Soendergaard et al. 2020 [68]	Higher serum IL6 Higher serum CD40L Higher serum osteocalcin	Nonresponse to vedolizumab in IBD patients Nonresponse to vedolizumab in CD patients Response to vedolizumab in UC patients
Bertani et al. 2020 [69]	Higher serum IL6 and IL8, more pronounced decrease of IL6 after 10 weeks	Clinical response to vedolizumab after 12 months
Battat et al. 2019 [70]	Increase of serum α4β7 and decrease of serum MAdCAM-1, VCAM-1, ICAM-1 and TNF	Clinical and endoscopic remission ate week 26 in vedolizumab-treated patients
Holmer et al. 2020 [71]	Higher serum VCAM-1 and ICAM-1 and lower serum α4β7	Endoscopic remission in vedolizumab-treated patients
Sands et al. 2017 [72]	Higher serum IL22	Clinical response to brazikumab

TREM-1: triggering receptor expressed on myeloid cells 1; CCR2-CCL7: chemokine receptor type 2–chemokine ligand 7; TNF: tumor necrosis factor; mTNF: membrane tumor necrosis factor; PBMC: peripheral blood mononuclear cells; LPS: lipopolysaccharide; OSM: oncostatin M; NK cells: natural killer cells; IL: interleukin; CD40L: ligand of cluster of differentiation 40; IBD: inflammatory bowel disease; CD: Crohn’s disease; UC: ulcerative colitis; MAdCAM-1: mucosal vascular addressin cell adhesion molecule 1; VCAM-1: vascular cell adhesion molecule 1; ICAM-1: intercellular adhesion molecule 1.

**Table 3 jcm-10-00853-t003:** Therapeutic drug monitoring.

Study	Cut-Off	Outcomes
**Anti-TNF Agents**		
Adedokun et al. 2014 [82]	Infliximab TLs ≥41 μg/mL at week 8 Infliximab TLs ≥3.7 μg/mL during maintenance	Clinical response, clinical remission, mucosa healing in UC patients Clinical response, clinical remission, mucosa healing in UC patients
Yarur et al. 2015 [83]	Infliximab TLs ≥8.3 μg/mL during maintenance	Mucosal healing in IBD patients
Roblin et al. 2014 [84]	Adalimumab TLs ≥6 μg/mL during maintenance Adalimumab TLs ≥6.5 μg/mL during maintenance	Clinical remission in IBD patients Mucosal healing in IBD patients
Bodini et al. 2016 [85]	Adalimumab TLs ≥10.1 μg/mL at week 48	Clinical remission in CD patients
Paul et al. 2013 [86]	Increase of infliximab TLs >0.5 μg/mL after dose-escalation	Mucosal healing in IBD patients
Papamichael et al. 2018 [78]	Infliximab TLs ≥7.5 μg/mL during maintenance Infliximab TLs ≥7.5 μg/mL during maintenance	Endoscopic healing in CD patients Histologic healing in CD patients
Kennedy et al. 2019 [87]	Infliximab TLs >7 μg/mL at week 14 Adalimumab TLs >12 μg/mL at week 14	Clinical remission at weeks 14 and 54 in CD patients Clinical remission at weeks 14 and 54 in CD patients
Adedokun et al. 2017 [88]	Golimumab TLs ≥2.5 μg/mL at week 6 Golimumab levels (steady-state) ≥1.4 μg/mL at week 44	Clinical response at week 6 in UC patients Clinical remission at week 54 in UC patients
Samaan et al. 2020 [89]	Golimumab TLs ≥3.8 μg/mL at week 6 Golimumab TLs ≥2.4 μg/mL during maintenance	Combined clinical and biochemical remission at week 6 in UC patients Combined clinical and biochemical remission during maintenance in UC patients
Magro et al. 2019 [90]	Golimumab TLs ≥2.9 μg/mL at week 6	Higher rates of clinical response, lower rates of endoscopic and histologic activity at week 6 in UC patients
Boland et al. 2019 [91]	Golimumab TLs ≥8.0 μg/mL during maintenance	Mucosal healing in CD patients during maintenance
Dreesen et al. 2019 [92]	Golimumab TLs ≥7.4 and 3.4 μg/mL at weeks 6 and 14	Endoscopic remission at week 14 in UC patients
**Vedolizumab**		
Rosario et al. 2017 [101]	Median vedolizumab TLs 26.8 μg/mL at week 6 Median vedolizumab TLs 34.7 μg/mL at week 6	Clinical remission at week 52 in CD patients Clinical remission at week 52 in UC patients
Osterman et al. 2019 [102]	Vedolizumab TLs ≥37.1 μg/mL at week 6, ≥18.4 μg/mL at week 14 and ≥12.7 μg/mL during maintenance	Clinical remission at week 52 in UC patients
Guidi et al. 2019 [103]	Vedolizumab TLs ≥16.55 μg/mL at week 14	Vedolizumab persistence in IBD patients
Yacoub et al. 2018 [104]	Vedolizumab TLs ≥18 μg/mL at week 6	Mucosal healing within the first year in IBD patients
Dreesen et al. 2018 [105]	Vedolizumab TLs ≥28.9 μg/mL at week 2 Vedolizumab TLs ≥23.4 μg/mL at week 6 Vedolizumab TLs ≥13.9 μg/mL at week 14 Vedolizumab TLs ≥13.5 μg/mL at week 22	Mucosal healing at week 14 in UC patients Biochemical remission at week 6 in CD patients Mucosal healing at week 14 in UC patients Mucosal healing at week 22 in CD patients
	Vedolizumab TLs ≥20.9 μg/mL at week 6 and ≥10.1 μg/mL at week 14 Vedolizumab TLs ≥26.2 μg/mL at week 6 and ≥30.1 μg/mL at week 14	Endoscopic improvement at week 14 Endoscopic remission at 6 months
Pouillon et al. 2019 [106]	Vedolizumab TLs ≥25 μg/mL during maintenance	Endoscopic and histological healing in UC patients
Miller et al. 2020 [107]	Vedolizumab TLs ≥27 μg/mL during maintenance	Clinical remission at week 52 in IBD patients
Vaughn et al. 2020 [108]	Vedolizumab TLs <7.4 μg/mL before dose-escalation	Response to dose-escalation in IBD patients
**Ustekinumab**		
Adedokun et al. 2018 [109]	Ustekinumab TLs ≥3.3 μg/mL at week 8 Ustekinumab TLs 0.8–1.4 μg/mL during maintenance	Clinical remission at week 8 in CD patients Clinical remission during maintenance in CD patients
Adedokun et al. 2020 [110]	Ustekinumab TLs ≥3.7 μg/mL at week 8 Ustekinumab TLs 1.1–1.3 μg/mL during maintenance	Clinical remission at week 8 in UC patients Clinical remission at week 44 in UC patients
Verstockt et al. 2019 [111]	Ustekinumab TLs ≥21.8 μg/mL at week 4 and ≥6.6 μg/mL at week 8 Ustekinumab TLs ≥2.7 μg/mL at week 16 and ≥1.9 μg/mL at week 24	50% decrease of faecal calprotectin in CD patients Endoscopic response at week 24
Soufflet et al. 2019 [112]	Ustekinumab TLs ≥2 μg/mL at week 8	Steroid-free clinical remission and biochemical remission at week 16
Battat et al. 2017 [113]	Ustekinumab TLs ≥4.5 μg/mL at week 26	Endoscopic response from week 26
Liefferinckx et al. 2020 [114]	Median ustekinumab TLs 2.45 μg/mL at week 16	Need for optimization during maintenance

CD: Crohn’s disease; IBD: inflammatory bowel disease; TLs: trough levels; TNF: tumor necrosis factor; UC: ulcerative colitis.

**Table 4 jcm-10-00853-t004:** Microbiological markers.

Study	Microbial Markers	Outcomes
Magnusson et al. 2016 [131]	Lower dysbiosis indexes and higher abundance of *F. prausnitizii*	Clinical response to anti-TNF in IBD patients Clinical response to anti-TNF in IBD patients
Shaw et al. 2016 [132]	Difference in abundance of specific genera (*Akkermansia, Coprococcus, Fusobacterium, Veillonella, Faecalibacterium,* and *Adlercreutzia*)	Clinical response to anti-TNF in IBD patients Clinical response to anti-TNF in IBD patients
Zhuang et al. 2020 [133]	Increased proportions of Lachnospiraceae and Blautia taxa at week 6	Clinical and endoscopic response to infliximab in CD patients
Wang et al. 2018 [134]	Higher abundance of SCFA-producing bacteria	Sustained response to infliximab in CD patients
Aden et al. 2019 [135]	Reduced metabolic interactions Higher levels of SCAFs after anti-TNF initiation	Non-response to anti-TNF in IBD patients Clinical remission in IBD patients
Seong et al. 2020 [136]	Increased bacterial diversity, richness and relative abundance of *F. prausnitzii*	Infliximab TLs >5 μg/mL at week 8 and mucosal healing within 3 months in IBD patients
Ananthakrishnan et al. 2017 [137]	Higher α-diversity, higher abundance of *Roseburia inulinivorans* and of a Burkholderiales species Metabolic pathways associated with microbial functions	Clinical remission in IBD patients treated with vedolizumab Clinical remission in IBD patients treated with vedolizumab
Doherty et al. 2018 [138]	Higher diversity and higher abundance of 2 OTUs affiliated with *Faecalibacterium* and *Bacteroides*	Clinical remission at week 6 in CD patients treated with ustekinumab

CD: Crohn’s disease; IBD: inflammatory bowel disease; OTU: operational taxonomic unit; SCFAs: short-chain fatty acids; TLs: trough levels; TNF: tumor necrosis factor; UC: ulcerative colitis.

## Data Availability

No new data were created or analyzed in this study. Data sharing is not applicable to this article.

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
