# Peer review of "Predictors and Early Markers of Response to Biological Therapies in Inflammatory Bowel Diseases"

_jcm, 2021, doi:10.3390/jcm10040853_

Round 1

Reviewer 1 Report

While I understand that the authors summarized present status of predictors and early markers of response to biological therapies in Inflammatory Bowel Diseases, all the contents descriptive, and the statements are not supported by statistical analyses.  As a consequence, the authors have failed to propose practical treatment strategy.  This seems to be a serious weakness of this review article. 

Author Response

We thank the reviewer for her/his comment. Indeed, we agree that the value of literature revision is greatly increased by applying statistical methods and meta-analyzing the results. However, this was beyond the scope of our present work, and we acknowledged this limitation by calling our paper both in the abstract and in the introduction a "narrative review".

Reviewer 2 Report

In this work, Privitera and colleagues provide a comprehensive review on predictors of respose towards biological therapy in patients with inflammatory bowel disease. This state-of-the-art review provides an in-depth presentation of genetic, immunologic, microbiological and traditional markers and their associations with disease response and it is very well and logically structured. Further, the authors have a very strong record in the field and this review will make an important contribution to the field.

Author Response

We would like to thank the reviewer for his or her time and effert spent in reviewing our manuscript and for the kind comment to our work.

Reviewer 3 Report

In this review the authors summarize the current knowledge of predictors and early markers of response to biological therapy.

In an era where personalized medicine seems to play an increasingly important role, I highlight this review for its clarity and organization.

Author Response

We would like to thank the reviewer for his or her time and effort spent in reviewing our manuscript and for the kind comment to our work.

Reviewer 4 Report

Thank you for the invitation to comment on this review by Privitera and colleagues. They address a range of clinically relevant and particularly current areas of IBD medicine. The review is comprehensive and summarises the wealth of available, and often conflicting, evidence without being overly exhaustive. I would have no issue with recommending publication but do have several linguistic/grammatical/stylistic comments as well as a couple relating to content, which I think should be addressed first.

Content:

  1. I like table 3 but think it really should, for completeness, include at least some data on goliumumab. There are data regarding optimal thresholds derived from post hoc analyses of PURSUIT (Adedokun et al. JCC 2017) as well as from several specifically designed prospective TDM studies (Samaan et al. AP&T 2020, Magro wt al. JCC 2019, Boland et al. IBD 2019, Dreesen et al. IBD 2019). These target a variety of endpoints that would fit well into table 3, along with a short accompanying passage in the text.
  2. I would be slightly careful about concluding that "the only tool of personalized medicine that has been incorporated in IBD clinical practice is represented by TDM for secondary non-response to infliximab and adalimumab". I would consider the use of TMPT quantification/genotyping and perhaps also HLADQA1*05 (though not yet widely available or fully validated) as tools of personalised medicine already widely used/soon to be incorporated into practice. I appreciate that they're not "Predictors and early markers of response to biological therapies in Inflammatory Bowel Diseases" but that isn't what the sentence says. Perhaps just make it biologic specific?
  3. "Identifying patients who are more likely to develop ADA would be of great help, as we know that concomitant immunosuppression (with thiopurines and, possibly, methotrexate) re- duces the risk of their formation." Why "possibly" methotrexate - PANTS clearly shows the benefit of MTX is entirely in keeping with thiopurines at preventing ADA formation. Would make this "(with thiopurines or methotrexate)".

Linguistic:

Intro

"sharp increase of incidence" should be "sharp increase in incidence"

"IBD carry a significant" should be "IBD carries a significant"

"when it comes to start therapies." should be "when it comes to starting therapies."

Immunosuppressants is a more common terms than "immunosuppressors"

Where anti IL-23 is mentioned, surely this should be 12 and 23? I presume you're talking about ustekinumab rather than the raft of p19's in trials.

"percentage of patients experiences secondary failures" should be "percentage of patients experience secondary failures"

"by a complicated disease" should be "by disease complications"

"point at the great necessity" should be "point to the great necessity" (could also lose the "great" from that sentence. "Pressing need" would be better?)

"specific therapy and to early assess their response" should be "specific therapy and to assess response at early point in treatment"

TDM section "infliximab; OR 0.13, 95% CI 0.06-0.28, pz0.0001 for adalimumab)" - presume the the z should be <

"Indeed, it is of the uttermost importance that predictive markers and models need to be validated in external cohorts, so as to prove their actual validity." uttermost should be utmost and the word actual probably isn't necessary.

"Comprehensively, the implementation of personalized medicine represents one of the most crucial unmet needs in IBD." The word comprehensively is misplaced here. Perhaps just say "To conclude" or "In conclusion"

Author Response

We would like to thank the reviewer for his or her comments. We have taken in considerations the suggestions and corrected the manuscript accordingly. Please, find a point-by-point answer to the comments. 

I like table 3 but think it really should, for completeness, include at least some data on goliumumab. There are data regarding optimal thresholds derived from post hoc analyses of PURSUIT (Adedokun et al. JCC 2017) as well as from several specifically designed prospective TDM studies (Samaan et al. AP&T 2020, Magro wt al. JCC 2019, Boland et al. IBD 2019, Dreesen et al. IBD 2019). These target a variety of endpoints that would fit well into table 3, along with a short accompanying passage in the text.

Author's reply: Thank you for your suggestions. We have included in table 3 data on golimumab. 

I would be slightly careful about concluding that "the only tool of personalized medicine that has been incorporated in IBD clinical practice is represented by TDM for secondary non-response to infliximab and adalimumab". I would consider the use of TMPT quantification/genotyping and perhaps also HLADQA1*05 (though not yet widely available or fully validated) as tools of personalised medicine already widely used/soon to be incorporated into practice. I appreciate that they're not "Predictors and early markers of response to biological therapies in Inflammatory Bowel Diseases" but that isn't what the sentence says. Perhaps just make it biologic specific?

Author's reply: We appreciate your comment and we agree with you. We have modified the manuscript accordingly. 

"Identifying patients who are more likely to develop ADA would be of great help, as we know that concomitant immunosuppression (with thiopurines and, possibly, methotrexate) re- duces the risk of their formation." Why "possibly" methotrexate - PANTS clearly shows the benefit of MTX is entirely in keeping with thiopurines at preventing ADA formation. Would make this "(with thiopurines or methotrexate)".

Author's reply: Thank you for your comment. We chose to be more prudent with methotrexate because evidence is less convincing. As you correctly mentioned, the PANTS study showed that methotrexate and thiopurines reduce the risk of ADA formation. On the other hand, the clinical benefit associated with methotrexate in combination with anti-TNF agents has not been clearly demonstrated: for instance, the COMMIT trial failed to meet its primary end-point and, similarly, a 2020 observational study by Targownkin and colleagues could not demonstrate a clear benefit associated with methotrexate-based combinations. However, we choose to follow your suggestion and to modify the sentence as suggested. 

Linguistic:

Intro

"sharp increase of incidence" should be "sharp increase in incidence"

Author's response: We have corrected the sentece as suggested.

"IBD carry a significant" should be "IBD carries a significant"

Author's response: We have corrected the sentece as suggested.

"when it comes to start therapies." should be "when it comes to starting therapies."

Author's response: We have corrected the sentece as suggested.

Immunosuppressants is a more common terms than "immunosuppressors"

Author's response: We have corrected as suggested.

Where anti IL-23 is mentioned, surely this should be 12 and 23? I presume you're talking about ustekinumab rather than the raft of p19's in trials.

Author's response: Thank you for your suggestions. We have corrected the text. Only in one sentence we presented data on anti-p19 drug (Trasncrptomic markers paragraph) and we left anti-IL23. 

"percentage of patients experiences secondary failures" should be "percentage of patients experience secondary failures"

Author's response: We have corrected the sentece as suggested.

"by a complicated disease" should be "by disease complications"

Author's response: We have corrected the sentece as suggested.

"point at the great necessity" should be "point to the great necessity" (could also lose the "great" from that sentence. "Pressing need" would be better?)

Author's response: We have corrected the sentece as suggested.

"specific therapy and to early assess their response" should be "specific therapy and to assess response at early point in treatment"

Author's response: We have corrected the sentece as suggested.

TDM section "infliximab; OR 0.13, 95% CI 0.06-0.28, pz0.0001 for adalimumab)" - presume the the z should be <

Author's response: We have corrected the typo.

"Indeed, it is of the uttermost importance that predictive markers and models need to be validated in external cohorts, so as to prove their actual validity." uttermost should be utmost and the word actual probably isn't necessary.

Author's response: We have corrected the sentece as suggested.

"Comprehensively, the implementation of personalized medicine represents one of the most crucial unmet needs in IBD." The word comprehensively is misplaced here. Perhaps just say "To conclude" or "In conclusion"

Author's response: We have corrected the sentece as suggested.

Round 2

Reviewer 1 Report

I have no comment.

Author Response

We would like to thank the reviewer for his or her time and effort.